# Comparison of Energy Budget of Cockroach Nymph (Hemimetabolous) and Hornworm (Holometabolous) under Food Restriction

**DOI:** 10.3390/insects15010036

**Published:** 2024-01-06

**Authors:** Charles J. Green, Chen Hou

**Affiliations:** Department of Biological Sciences, Missouri University of Science and Technology, Rolla, MO 65409, USA; cjgmmc@mst.edu

**Keywords:** energy budget, life history, food restriction, hornworm, cockroach

## Abstract

**Simple Summary:**

Low food availability imposes an energy tradeoff between metabolism for somatic maintenance and growth in animals. It has been hypothesized that energy allocation strategies under food restriction depend on whether the length of an animal’s potential development stage is longer or shorter than that of the food scarcity period, i.e., long development stages prioritize metabolism over growth, and short ones are the opposite. This hypothesis was partially proven true in hornworms (larvae of *Manduca sexta)*, a holometabolous insect species with a short development stage of ~4 weeks, but it is not clear if it holds for other insect species with long development. To further test this hypothesis, here, we use orange head cockroaches (*Eublaberus posticus*), which usually need several months to reach adulthood. We found that hornworms and cockroaches, two species with different life histories, have opposite energy allocation strategies under food restrictions, which may affect their health maintenance and lifespan.

**Abstract:**

Animals with different life histories budget their intake energy differently when food availability is low. It has been shown previously that hornworm (larva of *Manduca sexta*), a holometabolous insect species with a short development stage, prioritizes growth at the price of metabolism under food restriction, but it is unclear how hemimetabolous insect species with a relatively long development period budget their intake energy under food scarcity. Here, we use orange head cockroaches (*Eublaberus posticus*) to investigate this question. We found that for both species under food restriction, rates of metabolism and growth were suppressed, but the degree of reduction was more severe in growth than that of metabolism for cockroaches. Under both free-feeding and food restriction conditions, hornworms allocated a larger fraction of assimilated energy to growth than to metabolism, and cockroaches were the opposite. More importantly, when food availability was low, the fraction of assimilated energy allocated to growth was reduced by 120% in cockroaches, and the energy from growth was channeled to compensate for the reduction in metabolism; but, the fraction of assimilated energy allocated to growth was only reduced by 14% in hornworms. These results suggest that, compared to hornworms, cockroaches prioritize metabolism over growth.

## 1. Introduction

Animals need to budget the energy from food to maximize their fitness. Many empirical and theoretical studies have been conducted for understanding the energy allocation strategy of growing animals. The basic energy budgets described in studies are similar [1,2,3,4,5,6,7]. During growth, in a unit time, the energy assimilated from food, *A*, is partitioned between the energy deposited in new biomass, *G*, which is proportional to growth rate, and metabolic energy, *B*, which is dissipated as heat, i.e., A=G+B. The metabolic energy is further partitioned between energy for activities, biosynthetic work, and maintaining existing biomass. When facing low food availability, species with different life histories prioritize these energetic compartments differently [8,9,10]. Under food restriction, mammals and birds significantly slow down or cease growth and keep their mass-specific metabolic rate at the same, or a slightly lower, level as their free-fed (ad libitum, AL) counterparts [9]. In contrast, holometabolic insect species with short development periods, such as hornworms (larvae of *Manduca sexta*), prioritize growth over metabolism [11]. Although both growth rate and metabolic rate are undermined at the beginning of food restrictions, as caterpillars grow, hornworms invest more and more energy in growth, until, gradually, the fraction of energy allocated to growth increases closer to the level of their free-fed counterparts [11].

Mammals and birds utilize such a strategy because metabolic energy is required to maintain body temperature. More importantly, the low-food-availability period is usually shorter than their lifespan and is relatively temporary for them, so they can resume growth after the period is over (compensatory growth [12,13,14]) if they allocate energy to metabolism and keep good health. This means that their potential developmental period is longer than the low-food-availability period. In contrast, hornworms must grow and reach a threshold size for successful pupation in a short time period, usually 3~4 weeks depending on the temperature [15,16], so their developmental period is limited. If maintenance is prioritized at the price of growth, they will fail to pupate. It is unclear how hemimetabolous insect species with a relatively long development period budget their intake energy under food scarcity. On one hand, unlike endotherms, they do not need to maintain body temperature, so they may not need to keep the metabolic rate at the same level when food is limited, but, on the other hand, due to their long lifespan and developmental period, they may survive through food scarcity by suppressing growth to then later resume it, so they can afford to channel energy that would have been allocated to growth to compensate for the potential loss in metabolic rate. In this paper, we use orange head cockroaches (*Eublaberus posticus*) to investigate this question. The nymphal period of orange head cockroaches is usually 3 to 4 months (our preliminary results), depending on food availability and temperature, and, therefore, makes it a good model for this study.

## 2. Materials and Methods

### 2.1. Animal Rearing and Food Supply

In the fall of 2023, 24 individual orange head cockroach nymphs were raised at 25 °C, with no light and 90% humidity. Animals were randomly separated into two cohorts with different food supply levels (see below), each consisting of 12 nymphs in the 2nd to 4th instar. Each nymph was reared in an individual transparent container, 7 cm in diameter and 6 cm in height, with coconut fiber as the substrate. Cohorts with two food treatments were reared during the same period in the same incubator. This way, environmentally induced differences in growth and metabolism between two food treatments within a temperature are eliminated. The cockroaches were fed rat chow (Mazuri Exotic Animal Nutrition, St. Louis, MO, USA, 23% crude protein, 6.5% crude fat, and 4.5% crude fiber) and had two levels of food supply: ad libitum (AL) and food restriction (FR). The AL cohort was fed freely with an unlimited food supply. The FR cohort was fed every Monday, and food was completely removed after 24 h so that the cockroaches had no access to food for 6 days until fed again on Monday. Each nymph was measured repeatedly every week (see details below) for 15 weeks. They did not reach adulthood before the experiment ended.

### 2.2. Measurement of Growth Rate

Body mass was measured every Monday before refilling food to the nearest 0.1 mg on a digital microbalance (Perkin-Elmer AD6, PerkinElmer, Waltham, MA, USA). The average daily body mass gain is, therefore, estimated as weekly mass gain divided by seven. The growth rate, in units of watts, is defined as the daily increment in body mass multiplied by the energy content of the body tissue, i.e., $G=\Delta m\times E_{\mathrm{tissue}}/86400$, where Δm, in unit of grams, is the increment in body mass during the 24 h period, and *E*_tissue_ is the energy content of tissue, which is about 7000 Joules/gram [17,18].

### 2.3. Measurement of Metabolic Rate

We used a similar method described in our previous publication to measure the metabolic rate of hornworm larvae [19]. Before refilling the food supply on Monday, the rate of CO_2_ production, ${\overset{•}{V}}_{CO_{2}}$, of each nymph was measured for seven to ten minutes every week, using a flow-through respirometry system with an incurrent flow measurement [20]. A CA-10 CO_2_ analyzer (Sable Systems International (SSI); Las Vegas, NV, USA) was calibrated before all trials using air running through a column of drierite/ascarite (II)/magnesium perchlorate. The analyzer was then spanned with a gas of known CO_2_ concentration (300 p.p.m. CO_2_ in air). Baselines were taken before, in between, and after each trial by running air scrubbed of water and CO_2_ through the system. The time interval for baseline between each nymph was set to 5 min. The flow rate of the scrubbed air was set at 30 mL min^−1^ using an SS-4 subsampler (SSI). This air was then sent to the nymph or baseline chamber. During the trials, temperature was controlled at 25 °C, using a PELT5 temperature controller (SSI) that housed the respirometry and baseline chambers. Respirometry chambers for individual nymphs were 60 cc syringe barrels fitted with rubber stoppers connected to intake and outlet tubing.

ExpeData-P software (SSI) was used to correct for the drift in CO_2_ concentration. The rates ${\overset{•}{V}}_{CO_{2}}$ were calculated as ${\overset{•}{V}}_{CO_{2}}=\mathrm{FR}\times [CO_{2}]/100$, where FR is the flow rate, and [CO_2_] is the concentration of CO_2_ in the respirometry chamber [20]. Each data point represents the average of the measurement taken during the time interval. The larval metabolic rate, *B,* in units of watts, was calculated as $B=20\times {\overset{•}{V}}_{CO_{2}}/60$, assuming that the respiratory quotient is about 0.8 [21,22].

### 2.4. Data Analysis and Statistics

Since the nymphs were confined in a small container, and their activity was limited, we assume that the sum of the growth rate (*G*) and the metabolic rate (*B*) is approximately the rate of assimilation rate (*A*), i.e., A=G+B. Data on metabolic rate were logarithm transformed, Log(B)=Log(a)+d×Log(m), and ordinary least square linear regression was used to estimate the scaling coefficients and exponents. Data analysis was performed in OriginPro 2023 (OriginLab Corporation, Northampton, MA, USA).

## 3. Results

### 3.1. Growth Rate

The growth rate of the AL cohort increased with body mass, as $G=0.0071M^{1.45}$ (R^2^ = 0.20, df = 96, *p* < 0.001) (Figure 1). Under food restriction, the growth rate was independent of body mass (R^2^ = 0.004, df = 88, *p* = 0.55) (Figure 1). As cockroaches grew, food restriction significantly reduced the growth rate of cockroaches. On average, the growth rate in the AL cohort was 142.9 ± 216.3 Joules/day (N = 98), and that in the FR cohort was 30.0 ± 86.8 Joules/day (N = 90), so the FR-induced reduction in growth was 79%.

### 3.2. Metabolic Rate

The metabolic rate of the AL cohort scales with body mass as $B=0.23M^{0.903}$ (R^2^ = 0.584, df = 102, *p* < 0.001) (Figure 2). Under food restriction, the metabolic rate weakly scaled with mass as $B=3.58M^{0.481}$ (R^2^ = 0.108, df = 89, *p* = 0.002) (Figure 2). The FR-induced reduction was not significant when cockroaches were small, as below 700 mg, there were almost no differences in metabolic rate between these two groups; the difference becomes more significant when body mass is larger than 700 mg. The metabolic rate of the FR cohort was 140% of the AL cohort at a body mass of 300 mg, then decreased to 70% when body mass was at 1600 mg. Within this body size range, the overall average FR-induced reduction in metabolism was 7.2%.

### 3.3. Assimilation Rate

The assimilation rate (approximately equal to the sum of growth rate and metabolic rate) of the AL cohort scales with body mass as $A=0.11M^{1.14}$ (R^2^ = 0.29, df = 70, *p* < 0.001) (Figure 3) and that of the FR cohort did not scale with body mass (R^2^ = 0.01, df = 88, *p* = 0.272) (Figure 3). The assimilation rate of FR cockroaches was 52%±19.2% of that of the AL cohort.

### 3.4. Comparing Energy Budgets of Cockroaches and Hornworms

In this section, we compare the energy allocation priorities of cockroaches and hornworms when food availability is low. Previously, we measured the rates of growth and metabolism of hornworms under AL and FR conditions at three ambient temperatures [11]. Here, we compare the hornworm data collected at 25 °C to the data of cockroaches. The assimilation rate of hornworms under FR is 55.6%±8.6% of that of AL. So, the degree of food restriction on hornworms is comparable to that of cockroaches. What we are interested in are the fractions of assimilated food that were allocated to metabolism and growth, i.e., the ratios B/A and G/A, respectively, under AL and FR conditions in both species. Data show that none of these ratios is significantly correlated with body mass (R^2^ < 0.1 and *p* > 0.03 for all of the curves) (Figure 4). For this reason, we use the mean values of these ratios to compare the energy budgets of these two species.

We found that hornworms allocated 12.4%±5.4% of the energy assimilated from food to metabolism and 87.6%±5.4% to growth under ad libitum conditions. Under FR, the fraction allocated to metabolism increased to 24.8%±9.6%, and the growth was reduced to 75.1%±9.6%. In cockroaches, the fraction allocated to metabolism was 59.7%±90.8% and that to growth was 40.3%±90.8% when fed ad libitum. Under FR, the fraction for metabolism increased to 107.96%±161% and that of growth decreased to −7.96%±161% (Figure 5).

## 4. Discussion

Cockroaches are aggregative, and their development and behavior are influenced by isolation. But, in this study, each nymph was reared alone. This is because we needed to track the individual growth trajectories and take measurements repeatedly. If reared in groups, this would require marking each of them individually, which would be very difficult, if not impossible, considering molting and other factors that would cause the labels to fall off. However, the purpose of this study is to investigate the alteration in energy budget under food restriction. Since the ad libitum and food-restricted animals were reared in the same environment, isolation would influence both ad libitum and food-restricted samples. There is no evidence that shows the interaction effect of isolation and food supply level, so we assume that the influence of isolation is minimal when we compare the energy budget of the ad libitum and food-restricted animals.

Comparing the energy budgets of cockroaches and hornworms, three differences stand out. First, under both AL and FR conditions, hornworms allocated a larger fraction of assimilated energy to growth than to metabolism, and cockroaches showed the opposite. Second, more interestingly, when food availability is low, cockroaches showed greatly delayed growth from 40.3% to −7.96% and channeled the available energy to metabolism. The reduction in energy allocated to growth in cockroaches is $\frac{40.3-\left(-7.96\right)}{40.3}=120\%$. But the fraction of assimilated energy allocated to growth was only slightly reduced in hornworms, from 87.6% to 75.1%, and the reduction was only 14%. Third, although the pattern is statistically insignificant, it can be seen in Figure 4 that during growth, as body mass increased, the FR hornworms allocated more and more energy to growth, but the FR cockroaches allocated less and less to growth.

We hypothesize that the difference in energy budget, at least partially, depends on the length of their developmental period. For hornworms, the developmental period is only 3–4 weeks. In such a short period, they must reach a threshold of body mass; otherwise, they cannot successfully pupate. Thus, keeping fast growth under FR at the cost of low maintenance would be favored by selection, because, this way, the hornworms can not only reach the required size to pupate but will also have a relatively large size for high fecundity despite the low scarcity [23]. Cockroaches, however, whose developmental stage lasts several months, take a different strategy. On one hand, unlike endotherms, they do not need to keep a high metabolic rate in order to maintain body temperature homeostasis; on the other hand, unlike hornworms, they can resume growth after the low-food supply period (longer potential developmental period) and, therefore, do not have to keep a high growth rate under FR. Our results suggest that under FR, cockroaches prioritize metabolism over growth, opposite to what was observed in hornworms. Although both metabolism and growth were suppressed by FR, the degree of reduction in metabolism (7.2%) was much less than that of growth (79%) (Figure 2 and Figure 3) in cockroaches. More importantly, in cockroaches, the fraction of assimilated energy allocation to growth is greatly reduced to compensate for the reduction in metabolism (Figure 5A), but in hornworms, this compensation is much less (Figure 5B).

The available data generally support the hypothesis that the length of the potential developmental period determines the energy budget under food restriction. Mammals and birds prioritize metabolism at the expense of growth under FR. The studies on rats by McCarter and his workers [24,25] showed that when FR starts, the mass-specific metabolic rate initially decreases in FR animals but quickly returns to the same level as the AL animals. The trend of changes in the metabolic rate of FR rats is opposite of what we have observed in FR hornworms. Studies on “growth efficiency” also support this hypothesis. This efficiency is defined as body mass gain per unit of food intake, and, therefore is equivalent to and can be converted to the proportion of assimilated energy allocated to growth, *S*/*F*. Naim et al. [26] found that the growth efficiency in rats decreases at the beginning of FR, then increases for a short period but, eventually, decreases, which is also opposite to what has been seen in FR hornworms.

A similar conclusion can be drawn from a few studies on birds, although these studies only reported either the FR-induced changes in growth efficiency or the changes in metabolic scaling powers; they did not report both. It was found that Japanese quail [27] and broiler chicken [28] lower their growth efficiency under FR. In alcid chicks, including tufted puffin, horned puffin, crested auklet, and parakeet auklet, FR increased the metabolic scaling power [29]. The same change has also been observed in Japanese quail [30]. In sand martin, the metabolic scaling power is the same in FR animals as in their AL counterparts [31]. In a study of song thrush chicks [32], although the scaling powers were not reported, the mass-corrected metabolic rate was found to be higher in the FR animal. Among the studies we have found on how bird chicks respond to food restriction, only in European shag was the metabolic scaling power found to be lower in the FR chicks [33]. Due to the lack of data on food assimilation rates in these studies, we cannot estimate the exact changes in the proportion of metabolism in FR animals. However, an increase in metabolic scaling power in FR animals suggests that the FR animals increase the energy allocation to metabolism as body mass increases, opposite to what has been shown in hornworms.

Most studies on ectothermic animals’ energy budget under low food supply focus on non-growing animals [34,35,36,37,38,39,40] or the growth and metabolism of a population, instead of individuals [41,42]. However, limited data on growing ectothermic animals support our hypothesis. A non-diapausing nematode species, *Caenorhabditis briggsae*, takes a strategy close to hornworms. *C*. *briggsae*’s larval stage is about five days, and they do not enter the dauer stage (a state of suspended development and lowered need for energy intake) when food resources are low [43]. Thus, the length of their development stage is similar to hornworms, and their energy budget under FR is also similar to hornworms. Schiemer [43] found that FR decreases the metabolic scaling power in *C*. *briggsae*, and the growth efficiency in FR *C*. *briggsae* keeps increasing during the larval stage, whereas that of AL *C*. *briggsae* decreases near the end of the larval stage. Similar changes in metabolism and growth were observed in hornworms here.

In contrast, the Indian stick insect (*Carausius morosus*), a hemimetabolic insect species, takes a strategy similar to cockroaches. The Indian stick insect has a long juvenile stage that lasts 3–8 months [44]. With the long juvenile stage, Indian stick insects can potentially resume growth after a low-food-supply period and, therefore, do not have to prioritize growth under FR. Roark and Bjorndal [44] showed that under FR, the metabolic rate was lowered, but the scaling power remained the same as the AL counterparts. The authors did not measure growth rate nor the proportions of assimilated energy allocated to growth and metabolism. However, we can roughly estimate from the data reported that the FR-induced reduction in metabolic rate was about 24% (from their Figure 3), but the reduction in body mass by FR was more than 50% (from Figure 2), indicating a priority of metabolism over growth, similar to what was observed in cockroaches.

The energy budget under food restriction affects animals’ health maintenance and explains why the lifespan of many species can be extended by FR [45]. During growth, as stated in the Introduction, the energy assimilated from food (*A*) is partitioned between the metabolic energy (*B*) and energy deposited in the new biomass (growth, *G*). The metabolic energy is further partitioned between energy for normal activities, biosynthesis, and maintaining existing biomass, including repairing cellular damage, error-checking, etc. The energy deposited in the new biomass is the accumulated energy content of new biomass, and the energy for biosynthesis is the metabolic work required to synthesize the new biomass, which corresponds to the indirect work of growth and is completely dissipated as heat, not conserved in stored biomass. These two compartments are proportional to each other. When animals are under FR, the total energy from food decreases. If the metabolism and level of activities remain roughly unchanged, as shown in many studies on mammals (see review in [9]), then the deposition in new biomass must be suppressed. As emphasized above, the energy deposition in new biomass is proportional to the energy for synthesizing new biomass. When there is not as much energy to deposit, there is much less biosynthesis work for the animals. With an unchanged metabolism, the decreased requirement for the syntheses of biomass means more energy for maintenance and extended lifespan. Thus, FR channels energy from biosynthesis work to health maintenance. This is the energetic explanation for the observation that FR extends lifespan in mammals [45,46,47,48].

It remains unclear if FR will extend the lifespan of holometabolic insects with relatively short developmental periods. Our results suggest that hornworms try to maximize growth at the expense of metabolism under FR, so, the indirect cost of growth is not suppressed as much as in endotherms. Thus, with suppressed *B* and not much suppressed *B*_syn_, the FR-induced increase in energy for maintenance in hornworms is not as much as that in endotherms. So, we predict that the effect of FR on lifespan extension in hornworms will not be as significant as that in endotherms. Also, as far as we know, there is no study on FR’s effects on cockroaches’ lifespan. The only similar study was performed on the Indian stick insect. Roark and Bjorndal [44] showed that FR failed to extend its lifespan, indicating that FR fails to channel energy from the indirect cost of growth to maintenance due to this strategy. This leads us to call for more comparative studies to test the hypothesis that with the same level of food restriction, species with longer developmental periods benefit more in terms of health maintenance and longevity than species with short development.

Another noteworthy result from this study is the variation in metabolic scaling power induced by food restriction. The metabolic rate increases with body mass, as all else is kept equal [49,50]. The increase can be described by an allometric equation, $B=B_{0}m^{b}$, where *B*_0_ is a normalization constant, *m* is the body mass, and *b* is the allometric scaling power, around 3/4 [2,22,50,51,52,53]. The metabolic theory of ecology (MTE) [52] views the 3/4 scaling power as a canonical value and proposes that this value stems from optimizing the space-filling structure of the resource transport network in organisms [54,55]. In MTE, the physical structure of the transport network is assumed to be the primary constraint on the 3/4 scaling power, and other factors, which may affect the scaling power, are ignored in the approximation. However, intra-specifically, the metabolic scaling power, *b*, often varies within a range, usually between 2/3 and 1 [4,6,56,57]. The variation is associated with various life history factors, such as growth [58,59,60,61,62] and reproduction [63], and behavioral factors, such as activity [57]. The metabolic scaling power is also largely influenced by the ecological factors, such as ambient temperature [64] and predator–prey interaction [64,65]. In a recent study, DeLong et al. [66] proposed that changes in the food uptake rate with population density cause variations in metabolic rate. The authors showed that metabolic rate is negatively correlated with population density and attributed the density-dependent metabolic rates to the competitive effects on foraging rates, combined with an activity response to accommodate the resource constraint induced by competition. Within the framework of MTE, our result suggests that, when environmental factors (e.g., food supply level) vary, the structure and dynamics of the transport network may change accordingly, and, therefore, the metabolic scaling may change too. It is perhaps because the insect transport network (mainly the tracheal system for oxygen delivery) is more plastic when the food supply level varies. To fully test this hypothesis, more anatomic data on the tracheal system of food-restricted larvae are required.

## Figures and Tables

**Figure 1 insects-15-00036-f001:**
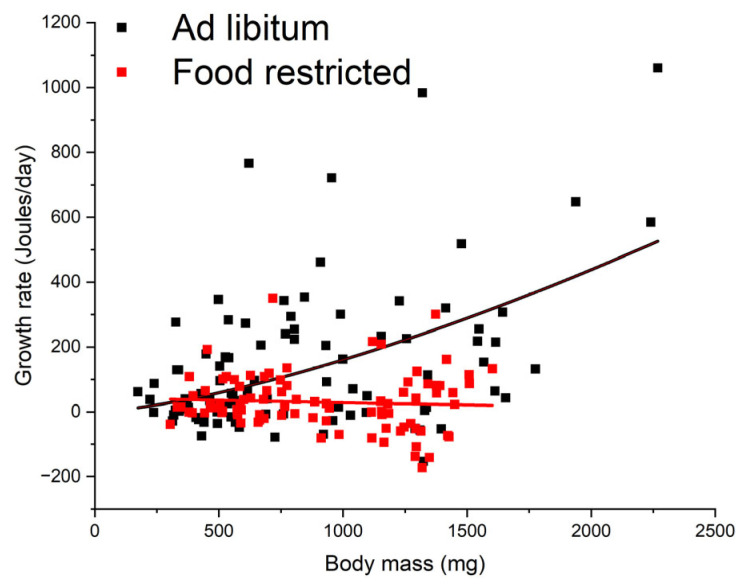
Growth rate versus body mass of the AL and FR cockroaches.

**Figure 2 insects-15-00036-f002:**
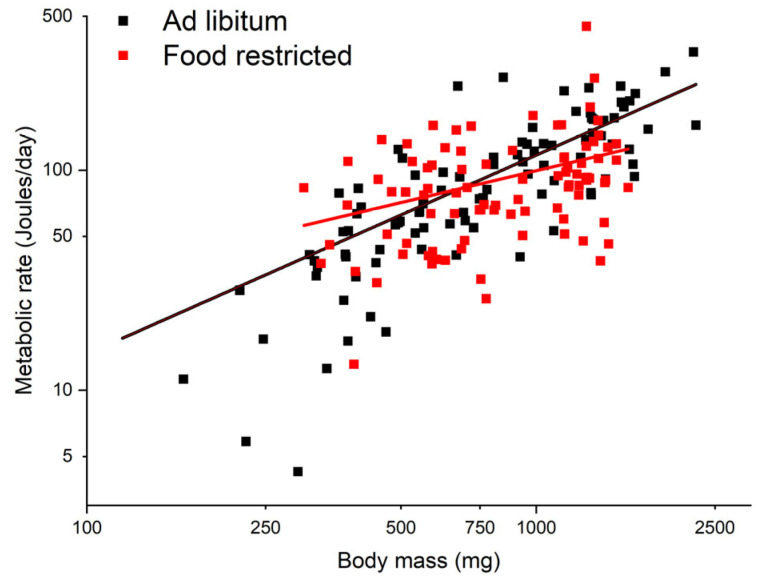
Metabolic rate versus body mass of the AL and FR cockroaches.

**Figure 3 insects-15-00036-f003:**
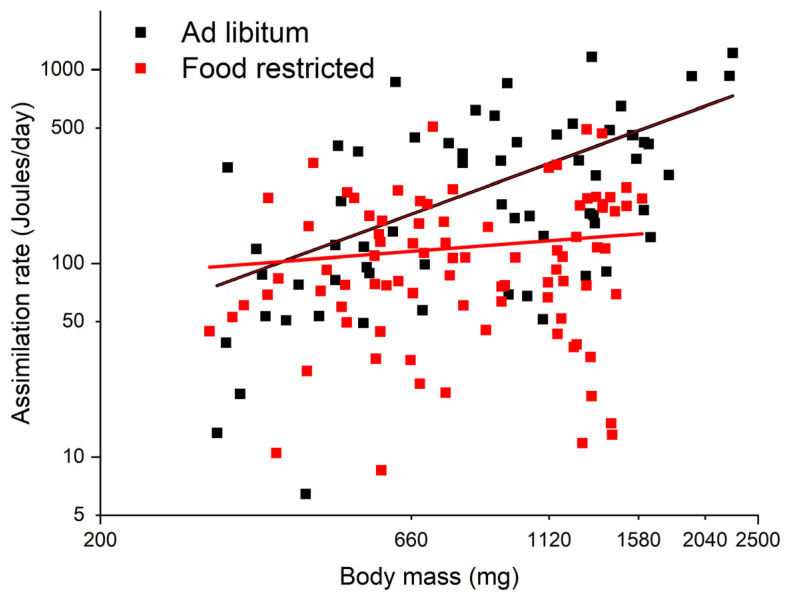
Assimilation rate versus body mass of the AL and FR cockroaches.

**Figure 4 insects-15-00036-f004:**
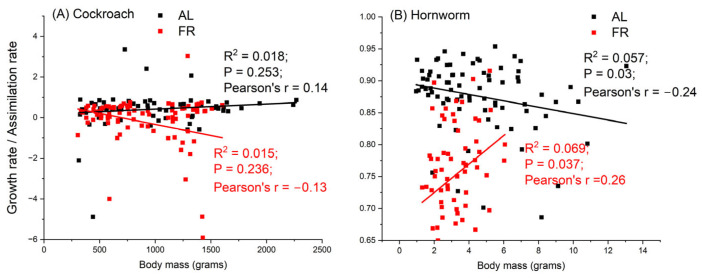
The ratios of energy allocated to growth to assimilated energy in cockroach (**A**) and hornworm (**B**) under ad libitum and food restriction over ontogeny. None of the ratios is significantly correlated with body mass.

**Figure 5 insects-15-00036-f005:**
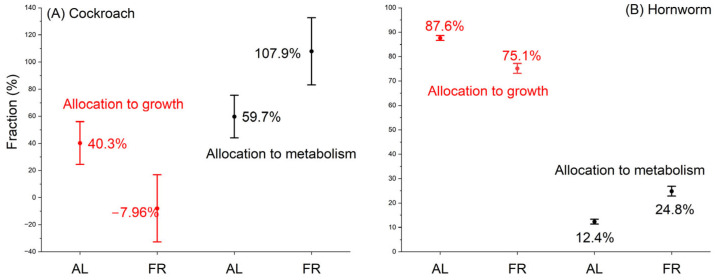
The fractions of energy allocated to growth and metabolism from the assimilated energy of cockroaches (**A**) and hornworms (**B**) under ad libitum and food restriction.

## Data Availability

The growth and metabolic rate data are available on DRYAD website: yed (accessed on 29 December 2023).

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
