# Peer review of "Comparison of Energy Budget of Cockroach Nymph (Hemimetabolous) and Hornworm (Holometabolous) under Food Restriction"

_insects, 2024, doi:10.3390/insects15010036_

Round 1

Reviewer 1 Report

Comments and Suggestions for Authors

In this M.S., the authors compare energy budget of orange head cockroaches (hemimetabolous) and hornworms (holometabolous) under food restriction condition. They found that for both species under food restriction, rates of metabolism and growth were suppressed, but the degree of reduction is more severe in growth than that of metabolism for cockroaches. Moreover, they show that cockroaches do not allocate a larger fraction of assimilated energy to growth than to metabolism under both free feeding and food restriction conditions. This finding has the potential to explain how hemimetabolous insect species budget their intake energy under food scarcity. I have just minor suggestions for improvement.  

1.     L78, it is unclear about the incubator conditions: eg. how many hours of light and darkness, 25°C? and relative humidity etc.

2.     L88, “at the same time”it is better to show the specific time. eg. What/Which day? and/or how many days before/after fed.

3.     Materials and Methods & L73: “24 Orange Head cockroach nymphs were raised…”. It is not clear if this is referring to technical replicates, or biological replicates. For this kind of study, it is important to be clear in reporting.

4.     L140. put a point “.” at the end sentence. 

5.     For all figures, x-axis: it is better to add “Body” as “Body mass (mg) or (g)”.

6.   I suggest to move L246-256 and L267-276 to Introduction part.

Author Response

We thank Reviewer 1 for all the suggestions, which help us to improve the paper. Below are our responses (in red font)

1. L78, it is unclear about the incubator conditions: eg. how many hours of light and darkness, 25°C? and relative humidity etc.

We have added the details as Reviewer 1 asked.

2. L88, “at the same time”,it is better to show the specific time. eg. What/Which day? and/or how many days before/after fed.

We have added: "every Monday before refilling food."

3. Materials and Methods & L73: “24 Orange Head cockroach nymphs were raised…”. It is not clear if this is referring to technical replicates, or biological replicates. For this kind of study, it is important to be clear in reporting.

We have added: "Each nymph was measured repeatedly every week (see details below) for 15 weeks. They did not reach adult before the experiment ended."

4. L140. put a point “.” at the end sentence. 

Added. Thank you for the careful reading.

5. For all figures,x-axis: it is better to add “Body” as “Body mass (mg) or (g)”.

Yes. Added. Thank you for the suggestion. 

6. I suggest to move L246-256 and L267-276 to Introduction part.

We hesitate to move them to the Introduction, because it would sound too abrupt. In the Introduction, we haven't talked about, in detail, how cockroach (and hornworm) budget their intake energy yet. So, if we compare them with other species, we are afraid that the readers may not know what we talk about. For this reason, we decided to keep these sentences unchanged. 

Reviewer 2 Report

Comments and Suggestions for Authors

Please see my comments in the attached file.

The title overstates the results of this study. There would need to be more representative species of both hemi- and holometabolous to support the title and assertions. I suggest a more modest title.

Cockroaches are aggregative, and their development and behavior are influenced by isolation. Was this taken into account? Why were the cockroaches held individually?

My primary concern is that the replicates described here are not real replicates. Individual cockroaches were measured repeatedly; thus, this is a repeated measures design, and the results and number of replicates need to be explained more carefully.

Are there any implications of this work for pest management?

Author Response

We really appreciate Reviewer 2's questions and suggestions. Below are our responses (in red font). 

The title overstates the results of this study. There would need to be more representative species of both hemi- and holometabolous to support the title and assertions. I suggest a more modest title.

Yes, we agree. We have changed the title to "Comparison of Energy Budget of Cockroach Nymph (Hemimetabolous) and Hornworm (Holometabolous) Under Food Restriction"

Cockroaches are aggregative, and their development and behavior are influenced by isolation. Was this taken into account? Why were the cockroaches held individually?

This is a very good question. Thank you for pointing it out. We have added a paragraph at the beginning of Discussion section to address it: 

"Cockroaches are aggregative, and their development and behavior are influenced by isolation. But in this study, each nymph was reared alone. This is because we needed to track the individual growth trajectories and take measurements repeatedly. If reared in groups, this would require marking each of them individually, which would be very difficult, if not impossible, considering molting and other factors that would cause the labels to fall off. However, the purpose of this study is to investigate the alteration in energy budget under food restriction. Since the ad libitum and food restricted animals were reared in the same environment, isolation would influence both ad libitum and food restricted samples. There is no evidence that shows the interaction effect of isolation and food supply level, so we assume that the influence of isolation is minimal when we compare the energy budget of the ad libitum and food restricted animals."

My primary concern is that the replicates described here are not real replicates. Individual cockroaches were measured repeatedly; thus, this is a repeated measures design, and the results and number of replicates need to be explained more carefully.

This is also an excellent point. We apologize for ignoring it in the original submission. We have explicitly stated it in the revision, and changed the statistic report. 

Are there any implications of this work for pest management?

Excellent question, thought-provoking. Thank you for that. Unfortunately, we are not experts in pest management, and we are no familiar with applied ecology, so we cannot offer profound and meaningful answers now. But, in our future research, we will keep this question in mind.

**************

 We have also made the minor changes, such as wording, brand name, food composition, etc, per Reviewer 2's suggestions in the manuscript. We really appreciate Reviewer 2's careful reading.